# Evaluation of Antifungal Activity of *Languas galangal* Rhizome and Development of a Topical Antifungal Cream

**DOI:** 10.3390/medicines10060034

**Published:** 2023-06-09

**Authors:** Lakshmi Reka, Chamari Maheshika Godage, Jayantha Wijayabandara, Aravinda Siriwardhene

**Affiliations:** Department of Pharmacy and Pharmaceutical Sciences, Faculty of Allied Health Sciences, University of Sri Jayewardenepura, Gangodawila, Nugegoda 10250, Sri Lanka; chamarimaheshika7@gmail.com (C.M.G.); siriwardhene@sjp.ac.lk (A.S.)

**Keywords:** antifungal activity, herbal cream, *Languas galangal* rhizome

## Abstract

**Background**: The rhizome of *Languas galangal* is traditionally used in Sri Lanka for the treatment of skin infections caused by fungi. The aim of the present study was to evaluate the antifungal activity of *L. galangal* rhizome and to develop a topical antifungal formulation from it. **Methods:** The dried, powdered rhizome of *L. galangal* was successively extracted with hexane, dichloromethane, ethyl acetate and methanol using Soxhlet extraction. The agar well diffusion method was used to assess the antifungal activity against *Candida albicans* and *Aspergillus nger*. The antifungal activities of the extracts were compared with clotrimazole as the positive control and dimethyl sulfoxide (DMSO) as the negative control. The most active hexane extract was used to prepare the cream. The antifungal activity of the formulated cream was tested. **Results:** The hexane extract of *L. galangal* rhizome powder was more effective on *C. albicans* and *A. niger*. The hexane extract of *L. galangal* showed the maximum zone of inhibition against *C. albicans* and *A. niger* (20.20 mm ± 0.46, 18.20 mm ± 0.46) compared to the other three extracts, while clotrimazole, which was used as a positive control, produced a larger zone of inhibition (36.10 mm ± 0.65) and dimethyl sulfoxide (DMSO), the negative control, did not produce inhibitory zones. Stability testing of the formulated cream showed a stable and good appearance. **Conclusions:** The cream developed using the hexane extract showed in vitro antifungal activity against *C. albicans* and *A. niger.* Further evaluations on shelf life, stability and safety are required.

## 1. Introduction

Mycosis is the term used to describe fungal infections in humans. Generally, fungal infections are chronic in nature. Superficial, cutaneous and, rarely, systemic fungal infections are prevalent in humans. *Candida albicans* and *Aspergillus niger* species are the most common fungal pathogens among the kingdom of fungi. Among them, *Candida* sp. is the most infective fungal species. Both can cause superficial infections of the ear, skin and nails. These infections are occasionally treated with topical antifungal therapy [1,2,3].

Plants and their extracts have immense potential for the treatment of fungal infections. Some of them have broad-spectrum antifungal activity [4]. Species of galangal plants are most widely studied because they have important medicinal properties [5]. *Languas galangal* is one of the most important species, belonging to the *Zingiberaceae* family. It is known by different synonyms such as “*Amomum galangal, Alpinia viridiflora, Maranta galangal, Languas galangal, Languas vulgare*”. *L. galangal* can be found in Indonesia, India, China, Saudi Arabia, Malaysia, Egypt and Sri Lanka and is known by several common names such as kaluwala in Sinhala; Kulanjan in Hindi; Arattha in Malayalam; Perarattai in Tamil; Mahabaracach, Sugandhavacha and Rasna in Sanskrit and greater galangal in English [6]. The rhizomes of *L. galangal* have a strong aromatic odor and a spicy or pungent taste, characterized by a dark reddish-brown color externally, and cuttings of the inner rhizome are identified by the presence of a dark center surrounded by a wider and paler layer on the outer rim [7,8].

Some studies were conducted to identify the antimicrobial activity of the rhizomes and the chemical constituents of different parts of *galangal* species. In the galangal rhizome, various compounds have been identified such as essential oils, flavonoids, phenolic acids, saponins and terpenoids [9,10]. Galangal acetate, kaempferol and 1,8-cineole are the main active compounds found in the galangal rhizome [11,12,13]. Diterpene compounds in some galangal species, such as -8β, 17-epoxylabd-12-ene-15 and 16-dial, play important roles in their antifungal activities against the growth of *C. albicans* [14,15]. The volatile oil of rhizomes with Sri Lankan origin has been analyzed using GC and GC/MS: sixteen compounds were identified, and the major constituent was zerumbone (44.8%) [15].

Although many investigations have been carried out to evaluate the antifungal activity of galangal species, antifungal dermatological preparations have hardly been developed from the plant. Therefore, scientific investigation and the preparation of a topical herbal cream from *Languas galangal* will help to develop novel treatments for diverse fungal infections. The main objective of the present work was to prepare and evaluate the antifungal activity of an herbal topical cream from *L. galangal* rhizome with Sri Lankan origin.

## 2. Materials and Methods

### 2.1. Chemicals and Reagents

The chemicals and reagents used in this study were hexane, dichloromethane, ethyl acetate (EtOAc), methanol (MeOH), 70% ethanol, sodium chloride solution, McFarland standard, sodium hydroxide (NaOH), hydrochloric acid (HCl), chloral hydrate, stearic acid, cetyl alcohol, triethanolamine, methylparaben, glycerin and liquid paraffin. Chloral hydrate and stearic acid were purchased from Daejung, Sheung-si, Korea. Cetyl alcohol and triethanolamine were purchased from Honeywell lab, Hessen, Germany. All other chemicals were purchased from Sigma Aldrich Co. Ltd., St. Louis, MO, USA. Aluminum-precoated silica gel plates 60 F254 (0.25 mm thick) used for analytical thin-layer chromatography (TLC) were purchased from Merck, Darmstadt, Germany.

### 2.2. Plant Material Collection

The plant *L. galangal* was collected from a home garden in Pilimathalawa, (latitude of 7.255° N 80.5188° E), Kandy district, in the Central Province of Sri Lanka, Kandy. The rhizomes of the plant were harvested in the last week of June in 2019, when the plant became six months old. Then, the rhizomes were separated from the stem and collected into dark-colored polythene bags. The plants were authenticated at the Bandaranayake Memorial Ayurvedic Research Institute, Navinna, Maharagama, Sri Lanka. The rhizomes were washed properly and cut into small pieces, air-dried and ground to a coarse powder and stored in air-tight bottles.

### 2.3. Macroscopic and Microscopic Studies

The macroscopic characteristics of the plants were observed properly by direct observation externally and by cutting the middle of the rhizome. Organoleptic evaluation was achieved by observing the shape, color, odor and appearance of the rhizome powder [16,17].

Microscopic studies were conducted by preparing a transverse section of the rhizome according to the standard method [18]. The rhizomes of *L. galangal* were washed with distilled water and then cut to obtain transverse sections in the watch glass. A free-hand, thinnest transverse section of rhizome was obtained. This transverse section was stained with safranin and then mounted on the slide. A few drops of chloral hydrate solution were added. The slide was covered with a glass cover slip and examined under the microscope, and different cell components were observed and recorded.

For powder microscopic studies, a few drops of chloral hydrate solution were added to a sample of powdered plant material on a slide and heated gently over a micro-Bunsen. The slide was covered with a glass cover slip for examination under the microscope, and different cell components were observed. The microscopic evaluation was supplemented with photomicrographs of different magnifications obtained using a Leica light microscope DM 1000 LED (Leica Microsystems CMS GmbH, Wetzlar, Germany).

### 2.4. Physicochemical Parameters

The percentages of the ash values were found according to the official methods described by *British pharmacopeia 2013* [19]. An empty silica crucible was heated for 30 min at a temperature of 600 °C and kept in a desiccator and weighed. About 2 g of the powder was accurately weighed in a tared silica crucible. The powder was spread as a thin layer at the bottom of the crucible. The crucible with the powder was incinerated at a temperature of 600 °C until it was free from carbon. The crucible with the powder was cooled and weighed. The procedure was repeated till a constant weight was observed. The percentage of the total ash was calculated with reference to the dried powder. The sample of total ash was divided into two equal samples to determine the percentage of acid-insoluble ash and water-insoluble ash.

Acid-insoluble ash: The ash was boiled for 5 min with 25 mL of dilute hydrochloric acid. The insoluble matter was collected in the crucible or on an ash-less filter paper and washed with hot water and ignited, and the weight percentage of acid-insoluble ash was calculated with reference to the air-dried drug.

Water-soluble ash: The ash obtained as described in the determination of total ash was boiled for 5 min with 25 mL of hot water. The insoluble matter was collected on an ash-less filter paper and washed with hot water. The insoluble ash was transferred into a tared silica crucible and ignited at a temperature of 600 °C. The procedure was repeated until a constant weight was observed. The weight of the insoluble matter was subtracted from the weight of the total ash. The difference in weight was considered as the amount of water-soluble ash. The percentage of water-soluble ash was calculated with reference to the dried powder.

The percentage loss on drying of plant materials was determined because any excess of water in raw material will encourage microbial growthThe weight of a freshly sliced rhizome was measured, and the sample was allowed to air dry under the fan for 24 h. Then, it was oven-dried at 40 °C for 6–8 h until there was no further loss of weight. The weight of the completely dried material was determined, and the percentage was calculated with reference to the initial weight.

### 2.5. Preparation of Extracts

A known amount (93.0 g) of powder was filled in the thimble of the Soxhlet extraction apparatus (Isolab, Wertheim, Germany), and it was refluxed successively with hexane, dichloromethane (DCM), ethyl acetate (EtOAc) and methanol (MeOH)350 mL of each solvent was used for the extraction. The extraction was performed in triplicate for three powder samples. The solvents of the extracts were evaporated using a rotary evaporator (model: RV 10 B, made by IKA, Staufen, German). These concentrated extracts were transferred to dried, cleaned glass containers and kept in the desiccator for removal of the excess solvents. Finally, the dried residues were stored at room temperature in air-tight containers for further investigations.

### 2.6. Evaluation of Antifungal Activity

An antifungal assay of the *L. galangal* extracts was conducted using the agar well diffusion method according to the National Committee for Clinical Laboratory Standards (NCCLS). Isolates of *C. albicans* (ATCC 25923) and *A. niger* (ATCC 16404) were obtained from the Department of Medical Laboratory Sciences, Faculty of Medical Sciences, University of Sri Jayewardenepura. Potato dextrose agar (PDA) was used as the culture medium for the agar well diffusion method. The final pH of the medium was adjusted to 5.2 ± 0.2 using a sodium phosphate buffer. *C. albicans* and *A. niger* plates were prepared for 3 days and 7 days, respectively [20]. These culture plates were used to prepare inoculums of *A. niger* and *C. albicans*. Two colonies of fungus were transferred to 5 mL 0.9% normal saline by using a sterile wire loop to gain equal turbidity for 0.5 McFarland standards (10^−6^ cfu/mL). Each fungal culture to be tested was spread on agar plates with a sterile swab moistened with the fungal suspension. Subsequently, the required number of wells were punched using a sterile pipette tip. The central well was filled with 20 μL of the positive control clotrimazole (5 mg/mL), and the remaining wells were loaded with 20 μL of 30 mg/mL hexane extract (H), 30 mg/mL DCM extract (D), 30 mg/mL EtOAc extract (E), 30 mg/mL MeOH extract (M) and negative control (5% DMSO) randomly. Then, the plates were left at room temperature for one hour to allow diffusion of the test samples and incubated for *C. albicans* and *A. niger* at 37 °C for 24 to 48 h [20]. After the incubation period, the diameters of all the zones of inhibition were measured. Then, the plant extract that yielded the highest zone of inhibition was selected for further investigations [21].

### 2.7. Developing a TLC Profile for the Selected Extract

TLC was performed for the most active extract of *L. galangal*. Firstly, a known weight (0.05 g) of the most active extract was dissolved in a known volume of hexane solvent (5 mL) to form a diluted hexane extract solution. About 10 μL of each extract was applied on precoated aluminum silica gel G 25 plates. Then, it was used to perform the TLC. The solvent system was developed to obtain the best resolution of the TLC chromatogram (Hexane:EtOAc, 8:2). Then, the developed TLC chromatograms were observed under 254 nm and 365 nm wavelengths using a fluorescence lamp. The retention factor (Rf) values of all the spots were determined by using the following formula: Retention factor = Distance traveled by the plant extract to the distance traveled by the solvent front.

### 2.8. Preparation of the Herbal Antifungal Cream from Hexane Extract

An oil-in-water (O/W)-based cream (20 g) was formulated. The emulsifier (stearic acid) and other oil-soluble components, thickening agent (cetyl alcohol) and emollient or lubricant (liquid paraffin) were dissolved in the oil phase and heated to 75 °C (Part A). the aqueous phase was prepared by dissolving the required amount of hexane extract in propylene glycol solvent and then adding it to water. The preservatives (methylparaben, triethanolamine) and the other water-soluble component glycerin, which was used as humectant, were dissolved in the aqueous phase and heated to 75 °C. After heating, the aqueous phase (Part B) was added in portions to the oil phase (Part A) with continuous stirring until cooling of emulsifier occurred [22]. The creation of the galangal extract cream was done according to the method mentionedin reference [22]. A 5% *w*/*w* cream was prepared using the formula given below (Table 1).

### 2.9. Antifungal Study of the Formulated Cream

Four different concentrations (10, 25, 50 and 100 mg/mL) were prepared from the formulated cream. Commercially available Clotrimazole cream was the positive control, and the cream base without active hexane extract used as the negative control. Both of these were dissolved separately in 5 mL of DMSO to fill in to the wells.After that, these test samples and the positive and negative control samples were filled into the wells of the prepared culture plate as previously mentioned.

### 2.10. Physical Evaluation

Physical assessments were carried out on the cream over a period of 14 days using organoleptic parameters such as color, odor, appearance and texture. The stability was determined by observing changes in the physicochemical parameters over 14 days.

The spreadability of the formulations was determined by measuring the spreading diameter of 0.1 g of sample between two horizontal glass plates after one minute, and the spreadability was compared with that of commercially available clotrimazole cream. A 0.5 g sample of the creams was dispersed in 50 mL of distilled water, and the pH was determined using a digital pH meter. Scarlet red dye was mixed with a small amount of cream and a drop of the cream was placed on a microscopic slide and covered with a cover slip; it was then examined under a microscope. It was monitored under a magnification of ×20, ×40 and ×100 through the microscope (Leica light microscope DM 1000 LED, Wetzlar, Germany), and microphotographs were obtained.

## 3. Results

### 3.1. Standardization of Raw Materials

#### 3.1.1. Macroscopic and Microscopic Characteristics

The aerial stem was a pseudo-stem consisting of a leaf sheath approximately 3.08 m in height. The leaves were simple, alternate, oblong, entire and acute at the base and apex of the plant with a very short petiole; the underside of the leaves was purple. The rhizome had a strong aromatic smell with conspicuous nodes and internodes. The rhizomes were characterized externally by a dark reddish-brown color, and cuttings of the inner rhizome were characterized by the presence of a dark center surrounded by a wider and paler layer on the outer rim (Figure 1).

A transverse section (TS) of the rhizome showed a layer of outer epidermis embedded with dark-brown pigments. The epidermis was multiple layered. It consisted of parenchymal cells with thin walls. The pith was very wide. It was circled with parenchymal cells embedded with starch grains and small intracellular spaces between the cells. Circular vascular bundles formed a continuous rim, and each vascular bundle was surrounded by a sclerenchymatous bundle sheath (Figure 2).

Microscopic powder analysis showed the presence of lignified phloem vessels, prisms of calcium oxalate crystals, starch grains, trichomes, long and short fibers, endocarps and oleo-resinous cells. The fibers were lignified with tapering ends. The phloem vessels displayed thickening. The starch grains were oval in shape and abundant. The trichomes were multicellular and thickened with sclereids (Figure 2).

Observations regarding the macroscopic and microscopic characters of the *L. galangal* confirmed that there is a similarity to early findings of these characteristic features of *L. galangal* in the eastern Himalayas and southwestern India [23].

#### 3.1.2. Physicochemical Parameters

Acid-insoluble ash mainly indicates contamination with earthy material. Water-soluble ash is used to estimate the quantity of the inorganic elements present in the drugs. The acid-insoluble ash of the sample indicated contamination with earthy materials in the sample. In all, 0.65% of earthy materials were present in the samples used in the present study. The water-soluble ash showed the amount of inorganic elements found in this study. The estimated water-soluble ash value of the sample in this study was 2.40%. Therefore, 2.40% of inorganic elements were present in the rhizome powder sample (Table 2). It is important to minimize adulterations to obtain a pure extract for study. The moisture content of the plant material used for the investigation was 10.35% *w*/*w*. The moisture content percentage of the plant rhizome was low. Therefore, it discouraged the growth of bacteria, fungi and yeast.

### 3.2. Phytochemical Screening

An examination of phytochemical properties of the galangal hexane extract was carried out including the tannin test, flavonoid test, saponin test, alkaloid test, steroid test and terpenoid test. The results of the phytochemical testing of the galangal hexane extract are shown in Table 3.

Based on Table 3, it can be stated that the galangal hexane extract contained tannins, flavonoids, alkaloids, terpenoids and steroids and did not contain saponins.

### 3.3. Thin-Layer Chromatography

The retention times for each chromatogram were observed as follows: Rf(7) = 0.9, Rf(6) = 0.85, Rf(5) = 0.725, Rf(4) = 0.625, Rf(3) = 0.55, Rf(2) = 0.45, Rf(1) = 0.3.

However, only one single spot was observed under 365 nm, which was a blue spot with an Rf value of 0.3. Therefore, it confirmed there are several bioactive compounds present in the hexane extract. In this study, TLC was performed under four different solvent systems, and the best separation was given by the Hexane:Ethyl acytate (8:2) solvent system.

The TLC results shows in Figure 3, indicates that the hexane galangal extract produced spots at Rf 0.62 and Rf 0.72. These findings related to the observations in reference [24]: Rf 0.62 shows the properties of a simple phenolic spectrum, and Rf 0.71 is phenyl propene. More spots corresponding to secondary metabolite compounds were isolated or separated from other components.

### 3.4. Antifungal Assay

The extracts of *L. galangal* were also compared against the standard antifungal agent clotrimazole. It was found that 5 mg/mL clotrimazole, used as a positive control, produced a larger zone of inhibition (36.10 mm ± 0.65). The hexane extract of *L. galangal* showed the maximum zone of inhibition against *C. albicans* and *A. niger* (20.20 mm ± 0.46, 18.20 mm ± 0.46) compared to the other three extracts. A moderate zone of inhibition was shown by the methanol extract. The antifungal activities of the crude extracts (hexane extract, DCM extract, EtOAc extract, MeOH extract) are given in Figure 4, and the results are summarized in Table 4.

Table 5 represents the diameter of the zone of inhibition for *C. albicans* and *A. niger* at different concentrations of the herbal cream. The zone of inhibition increased with the concentration of the herbal cream. Dose-dependent antifungal activity was observed. The maximum inhibition was seen against *C. albicans* (32.81 mm ± 0.21) at the highest concentration of herbal cream (100 mg/mL). The second largest zone of inhibition was against *A. niger* (28.84 mm ± 0.33) at the same concentration (100 mg/mL). No zone of inhibition was observed for the negative control. The lowest concentration of herbal cream (10 mg/mL) did not show inhibition against *A. niger*.

### 3.5. Physical Evaluation of the Formulated Creams

The microphotographs of the cream confirmed that the formulated cream was an oil in water type of emulsion cream. Disperse globules were seen, and the background was colorless. Therefore, it confirmed that the formulated cream is an oil-in-water type cream. The cream was slight yellowish in color due to the color of the extract. The pH of the formulated cream was found to be 5.1, which is a good and recommended pH for the skin. The spreading diameter of the formulation was compared with that of commercially produced clotrimazole cream. The hexane extract cream had more spreadability than clotrimazole cream in this study. The formulated antifungal cream was evaluated for several physicochemical characteristics, and the results are shown in Table 6. The results of the stability tests showed that there were no changes in the color, odor, spreadability, texture or pH, and no phase separation or coalescence was observed over the period of time.

## 4. Discussion

Standardization is an important process for herbal drugs to establish the identity, purity, safety and quality of the formulation to achieve the desired therapeutic effects [25,26]. According to the World Health Organization (WHO), the macroscopic and microscopic testing of a medicinal plant is the initial step toward determining the identity and the degree of purity of such material and should be achieved before any further tests are undertaken [27]. In order to standardize the drug, various macroscopic and microscopic physicochemical parameter and phytochemical analyses were carried out in the present study. Microscopic assessment of the plant material is important for the detection of source materials. This gives an idea of the different features of the plant material such as cork cells, cortex, secondary phloem and fibers [28].

The physicochemical parameters such as loss on drying and ash content are also vital for the standardization and quality control of herbal drugs. Loss on drying is the most common test used to determine the moisture content in a powdered sample. The moisture content of drugs should be at a minimal level to minimize the growth of bacteria, yeast or fungi. The ash value is a useful parameter to determine the quality and purity of a crude drug. It shows the presence of various impurities such as carbonate, oxalate and silicate. The acid-insoluble ash mainly consists of silica and indicates contamination with earthy material, and the water-soluble ash is used to estimate the amount of inorganic compounds present in drugs [29]. Furthermore, the World Health Organization (WHO) has declared that chromatographic fingerprinting is an important technique for the quality evaluation of medicinal herbs [30]. The TLC analysis in the present investigation was a non-targeted analysis, and the main aim was to obtain a satisfactory separation of all substances contained in the tested samples. Different mobile phases were tested in order to obtain reproducible peaks, and finally, the Hexane:Ethyl acetate (8:2) solvent system was chosen and a fingerprint profile was developed. The retention times for each chromatogram were observed as Rf(7) = 0.9, Rf(6) = 0.85, Rf(5) = 0.725, Rf(4) = 0.625, Rf(3) = 0.55, Rf(2) = 0.45, Rf(1) = 0.3. Therefore, it confirmed that there are several bioactive compounds present in the hexane extract. TLC profiling is the best method to reveal the various secondary metabolites for the rapid identification of proper raw materials without any adulteration.

Preliminary phytochemical screening can predict the different constituents present in solvents with different polarities [31]. These results will be useful to separate the pharmacologically active metabolites present in a plant for future investigations. The phytochemical study of *L. galangal* showed that it contained tannins, flavonoids, alkaloids, terpenoids and steroids but not saponins.

The aim of the current study was to develop an antifungal cream from *L. galangal* rhizome. Initially, this study planned to identify the best extract with the greatest antifungal activity. The antifungal activities of hexane, dichloromethane, ethyl acetate and methanol extracts of the *L. galangal* rhizome were tested against *C. albicans* and *A. niger.* The hexane extract of *L. galangal* showed the maximum zone of inhibition against *C. albicans* compared to the other three extracts. Moderate antifungal activity was shown by the methanol extract. Furthermore, each extract showed higher antifungal activity against *C. albicans* than against *A. niger*.

This significant result may be due to the presence of active chemical constituents in the plant extract. The most active hexane extract of the *L. galangal* rhizome was used to formulate a cream preparation, and it showed good antifungal activity. The maximum inhibition was shown against *C. albicans* (32.81 mm ± 0.21) at the highest concentration of the herbal cream (100 mg/mL). This shows that the formulation used in this study has good in vitro antifungal activity against *C. albicans* and *A. niger.* Although this cream showed good antifungal activity, further evaluation of its safety is required. According to the previously published research, the dermal application of undiluted galangal extract is not associated with any toxic effect on the non-abraded skin of rabbits [32]. Nevertheless, this study should be continued to check whether the CC50 is lower than the IC50 (50% inhibitory concentration for fungal growth).

Cream base types with a high viscosity will reduce the release of the drug from the base. Stearic acid was used at concentrations of 1–20%, and triethanolamine (TEA) was used to prevent irritation of the skin. Stearic acid reacts with TEA in situ to produce a salt, triethanolamine stearate, which functions as an emulgator for oil-in-water type emulsions [33,34]. Therefore, the above-mentioned formula was used to prepare an antifungal cream. The prepared antifungal cream showed a satisfactory level of physical characters such as homogeneity, spreadability and viscosity, and no phase separation, creaming, turbidity, cracking, coalescence or phase inversion was observed during stability tests.

## 5. Conclusions

The purpose of this study was to develop an herbal cream. The hexane extract of *L. galangal* was identified as the best extract with stronger antifungal activity against *C. albicans* and *A. niger* than the other three extracts. Furthermore, its effect remained even when it was included into the formulation of an antifungal cream. The results of different physicochemical tests showed a stable and good appearance of the cream. Further detailed stability and safety studies are needed to improve the overall quality of the product. The standardization and quality-control test results obtained from the present study will also be helpful in the preparation of a monograph. Furthermore, these parameters will be useful for the confirmation of the identity and authenticity of the *L. galangal* plant.

## Figures and Tables

**Figure 1 medicines-10-00034-f001:**
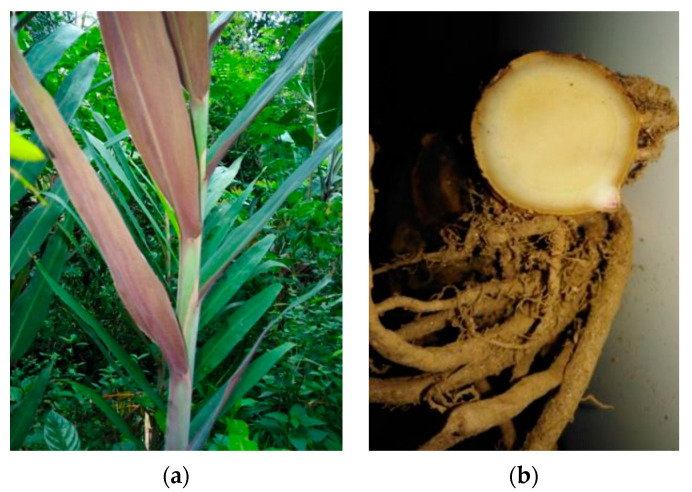
Macroscopic characteristics of *L. galangal.* (**a**) Morphological view of the *L. galangal.* (**b**) Cuttings of the inner rhizome.

**Figure 2 medicines-10-00034-f002:**
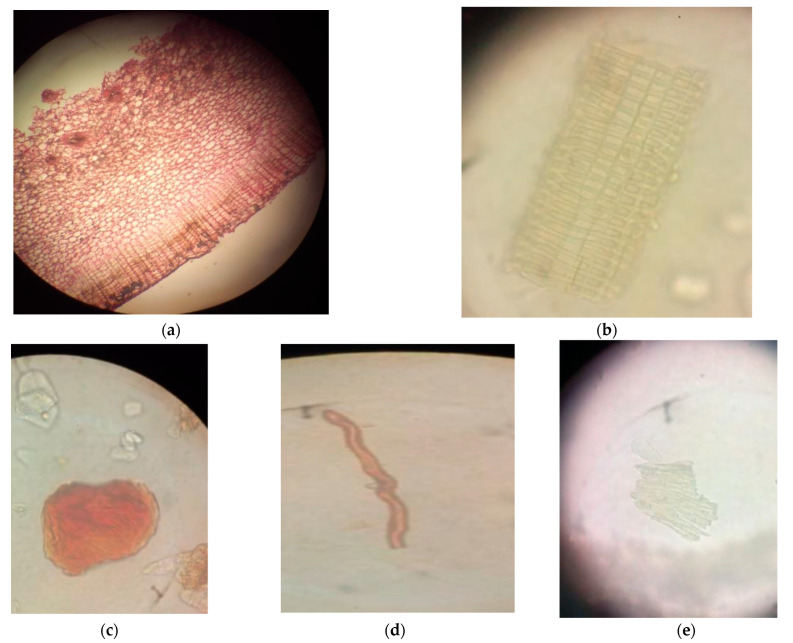
(**a**) Microscopic view of the transverse section of *L. galangal* rhizome (×40). Powder microscopy: (**b**) phloem vessel (×40), (**c**) calcium oxalate crystals (×40), (**d**) fiber (×40), (**e**) endocarp (×40).

**Figure 3 medicines-10-00034-f003:**
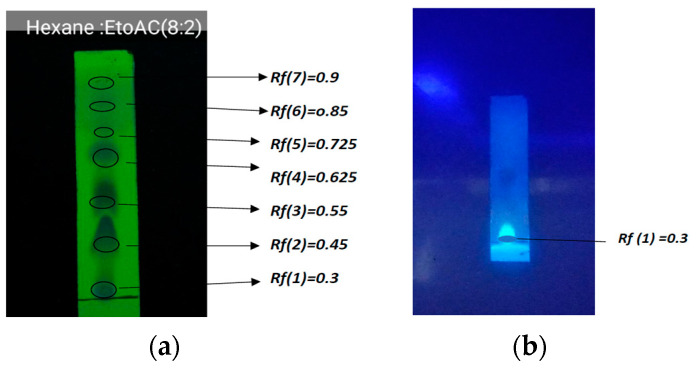
TLC results for the hexane galangal extract. (**a**) TLC galangal extract under the UV 254 lamp. (**b**) TLC under the UV lights, 360 nm.

**Figure 4 medicines-10-00034-f004:**
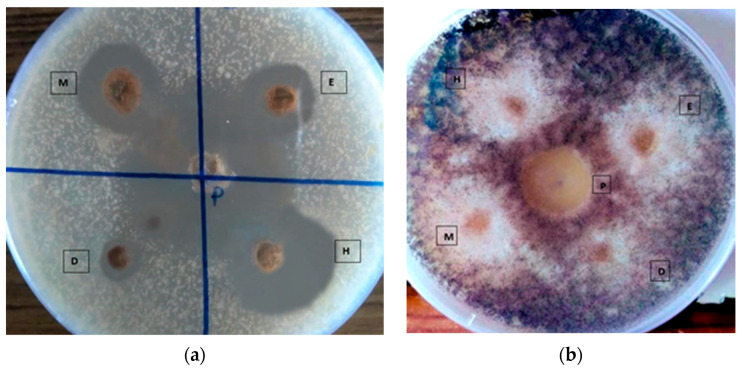
Antifungal activity of the extracts of *L. galangal* rhizome. (**a**) *Candida albicans.* (**b**) *Aspergillus niger*.

**Table 1 medicines-10-00034-t001:** Formulation and composition of antifungal cream.

Components	Ingredients	Amount in Grams (g) for 5% *w*/*w* Cream
	Hexane extract of *L. galangal*	1
Oil phase (Part A)	Stearic acid	2.2
	Cetyl alcohol	0.8
	Liquid paraffin	0.8
Aqueous phase (Part B)	Water	14.7
	Triethanolamine	0.3
	Glycerin	1.0
	Methylparaben	0.2

**Table 2 medicines-10-00034-t002:** Percentages of ash values and loss on drying.

Ash Values	Percentage
Total ash values	5%
Acid-insoluble ash	0.65%
Water-soluble ash	2.40%
Loss on drying, 110 °C	10.35%

**Table 3 medicines-10-00034-t003:** Phytochemical test results for galangal hexane extract.

Phytochemicals	Results
Flavonoids	+
Saponins	−
Tannins	+
Alkaloids	+
Steroids	+
Terpenoids	+

**Table 4 medicines-10-00034-t004:** Antifungal activity of the extracts of *L. galangal* rhizome.

Plant Extracts	Mean Diameter of Zone of Inhibition mm ± SE	
	*C. albicans*	*A. niger*
Positive control (P) (5 mg/mL clotrimazole)	36.10 ± 0.65	24.20 ± 0.32
Negative control (N) (5% DMSO)	No zone	No zone
Hexane extract (H) 30 mg/mL	20.20 ± 0.46	18.20 ± 0.46
Methanol extract (M) 30 mg/mL	17.30 ± 0.79	12.30 ± 0.72
Dichloromethane (D) 30 mg/mL	4.34 ± 0.80	4.30 ± 0.71
Ethyl acetate extract (E) 30 mg/mL	6.80 ± 0.54	6.70 ± 0.69

Source: experimental results.

**Table 5 medicines-10-00034-t005:** Antifungal activity of the hexane extract of *L. galangal* rhizome at different concentrations of formulated cream.

Different Concentrations mg/mL	Mean Diameter of Zone of Inhibition mm ± SE	
	*C. albicans*	*A. niger*
10	6.30 ± 0.24	Nil
25	10.20 ± 0.14	8.2 ± 0.31
50	28.40 ± 0.30	23.2 ± 0.21
100	32.81 ± 0.21	28.84 ± 0.33
Negative control (N) (Base only)	Nil	Nil
Positive control	38.88 ± 0.22	29.54 ± 0.32

Source: experimental results.

**Table 6 medicines-10-00034-t006:** Physicochemical evaluation of the formulated cream.

Parameter	Results
Homogenicity	Good
Appearance	Pale yellow
Odor	Good (aromatic)
Spreadability	Good
Texture	Emollient and slippery
Type of smear	Non-greasy
Removal	Easy
Stability	Stable
pH	5.1

Source: experimental results.

## Data Availability

The data used to support the findings of this study are available from the corresponding author upon request.

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
