# Peer review of "Evaluation of Antifungal Activity of Languas galangal Rhizome and Development of a Topical Antifungal Cream"

_medicines, 2023, doi:10.3390/medicines10060034_

Round 1
Reviewer 1 Report
The current manuscript is an interesting study on the antifungal activity of Languas galangal rhizome, with the development of a topical cream. Although several studies were performed, the manuscript lacks information in what concerns some detailed protocols and equipment, and some discussion in lacking. Also, grammatical, organization and punctuation errors should be corrected. Authors should follow these suggestions for manuscript improvement:
- Lines 38 to 42 have wrong use of italics, should be corrected;
- Throughout the manuscript, sometimes there are spaces missing between phrases or references and words, such as in line 50 or line 45, this should be checked and corrected;
- Some titles end with a full stop mark “.”, this should be removed;
- Microscope type, brand and city + country of origin should be added in the methods section;
- The reference for “British pharmacopeia 2013” should be added; in that same section “2.3. Physico-chemical analysis” the method should be further described, including the used equipment and specific protocol;
- In the chemicals and reagents section the brand, name and city of each reagent should be provided;
- Why was this specific cream formula chosen? Authors should justify the choice; also the function of each excipient in the formulation should be provided;
- In section 2.10 says “Microphotographs were taken”, again information about the methods and apparatus should be provided;
- Figures and Tables should be placed right after they are first mentioned in the text;
- Results should not only be stated, but should also be critically discussed (not just in the final discussion section).
Grammatical, organization and punctuation errors should be corrected.
Reviewer 2 Report
The paper is generally well written and structured. I strongly encourage the authors to address the following points
Line 30 – Author suggested to expand the genus name of the fungi when used at first.
Line 30 – “and” must be normal font.
Line 38 – 43 – must be normal font other than fungi genus and species name, if used.
Genus and species name must be abbreviated at first and follow the same format throughout the manuscript. For e.g. author written C. albicans in line 30 and Candida albicans in line 52.
2.1. Plant Material collection – how long the plant material has been collected? is there any seasonal changes will be there.
do rhizomes have light sensitivity as the rhizomes are grown under the soil?
Line 75 – “L. galangal” must be italic.
Line 79 – 84 – Check the sentence repetition. if it is not repetition, change the sentence and make it simple and clear.
2.3. Physico-chemical analysis – sentence is not clear.
Line 95 – the triplicate of powder samples is prepared in the same time or different time point.
Candida albicans (ATCC 25923) and Aspergillus niger (ATCC 16404) – is there any specific passage number available?
Relocate 2.5. Chemicals and reagents section – at the beginning of materials and methods.
Line 112 - “Candida albicans and Aspergillus niger plates were prepared 3 days and 7 days respectively”. what is the reason for choosing 3 days and 7 days?
Line 118 - positive control clotrimazole concentration?
Line 119 - How the author chooses 30mg/ml concentration?
Line 122 - thirty milligrams (30mg) what it means and Concentration of DMSO?
Line 125 - 24 to 48 hours, reason, or reference for time selection?
Sentence need correction – “TLC was performed for the most active extract of Languas galangal. Firstly, the known weight (0.05g) of most active extract was dissolved in known volume of hexane solvent (5ml) to form diluted hexane extract solution” – in 2.6. Evaluation of antifungal activity section author mentioned 30mg/ml but here they used 0.05g, - its making confusion.
In the Table 1. what it means of g & F1?
Line 150 – “Four” f must be in small letter.
Line 159 – double check the format of witting pattern of °C
Line 201 – figure legend is must be visible.
Line 223 – units must be at the end of the values.
Summary of Table 4 & Figure 4.– extract concentration? treatment time point?
Line 233 – “The zone of inhibition did increase with higher concentration”. – sentence not clear.
Line 240 – remove . between “rhizome.at”.
Line – 265 – Sentence need clarity – “Furthermore, all four extracts show higher activity against C. albicans”
Discussion is very short; author need to discuss their work with more references to improve the quality of the manuscript.
Author suggested to double check the genus and species names are in the same format including italic, space between two words, full stop, comma, open and close the brackets, everything throughout the manuscript.
